# Mouse Cytomegalovirus Differentially Exploits Cell Surface Glycosaminoglycans in a Cell Type-Dependent and MCK-2-Independent Manner

**DOI:** 10.3390/v12010031

**Published:** 2019-12-27

**Authors:** Sergio M Pontejo, Philip M Murphy

**Affiliations:** Molecular Signaling Section, Laboratory of Molecular Immunology, National Institute of Allergy and Infectious Diseases, National Institutes of Health, Bethesda, MD 20892, USA; pmm@nih.gov

**Keywords:** cytomegalovirus, HCMV, MCMV, viral cell entry, glycosaminoglycans, heparan sulfate, chondroitin sulfate, viral chemokines, MCK-2

## Abstract

Many viruses initiate interaction with target cells by binding to cell surface glycosaminoglycans (GAGs). Heparan sulfate (HS) appears to be particularly important in fibroblasts, epithelial cells and endothelial cells, where it represents the dominant GAG. How GAGs influence viral infectivity in HS-poor target cells such as macrophages has not been clearly defined. Here, we show that mouse cytomegalovirus (MCMV) targets HS in susceptible fibroblasts and cultured salivary gland acinar cells (SGACs), but not in macrophage cell lines and primary bone marrow-derived macrophages, where chondroitin sulfate was the dominant virus-binding GAG. MCK-2, an MCMV-encoded GAG-binding chemokine that promotes infection of macrophages as part of a gH/gL/MCK-2 entry complex, was dispensable for MCMV attachment to the cell surface and for direct infection of SGACs. Thus, MCMV tropism for target cells is markedly influenced by differential GAG expression, suggesting that the specificity of anti-GAG peptides now under development as HCMV therapeutics may need to be broadened for effective application as anti-viral agents.

## 1. Introduction

Human cytomegalovirus (HCMV) is the largest member of the *Herpesviridae* family and chronically infects ~60–80% of adults [1]. Infection is usually asymptomatic in healthy individuals but can cause severe pathology, including retinitis, encephalitis, pneumonia and hepatitis, in immunocompromised hosts [2]. HCMV is also the leading infectious cause of fetal morbidity, which imposes a major economic burden estimated at ~$2 billion/year in the USA [3]. Accordingly, the US National Academy of Medicine designated HCMV as a top public health priority in 1999 [4].

HCMV vaccine strategies have focused mainly on blocking cell entry by the virus. A generally accepted entry mechanism involves the binding of HCMV glycoprotein gH/gL complexes to cellular receptors, which triggers a conformational change in the HCMV gB protein, thereby inducing fusion of the viral envelope with the target cell membrane [5]. In HCMV, two different gH/gL complexes have been identified—the trimer gH/gL/gO, which is required for HCMV infection in most cell types [6,7], and the pentamer gH/gL/UL128-UL130-UL131A, which is dispensable for fibroblast infection but essential for infection in leukocytes, endothelial cells and epithelial cells [8,9,10]. Interestingly, UL128 and UL130 are HCMV-encoded homologues of host chemokines [11,12], a large family of cytokines that coordinate leukocyte trafficking by binding to 7-transmembrane domain receptors. However, no chemokine receptors have been identified for UL128 or UL130, and the two published studies reporting chemotactic activities for UL128 came to seemingly opposite conclusions [13,14]. Functional homologues of HCMV gB and gH/gL/gO have been found in its mouse counterpart, mouse cytomegalovirus (MCMV) [15,16]. However, in place of the HCMV pentamer, MCMV produces a gH/gL/MCK-2 complex required for infection of macrophages [17]. Like HCMV UL128 and UL130, MCMV MCK-2 is a viral chemokine, syntenic but only distantly related to HMCV pentamer chemokines (<20% amino acid identity) [18]. MCK-2 has been defined as an MCMV virulence factor since MCK-2-deficient viruses establish a lower viral burden in mouse salivary glands, a central reservoir for virus persistence and transmission [19,20]. In addition, recombinant MCK-2 has been reported to induce calcium flux in peritoneal cells, and to cause inflammation and myeloid cell recruitment when injected in the mouse footpad [21,22]. However, the molecular mechanism of the MCK-2-dependent infection of the salivary gland and the putative MCK-2 cellular receptor remain undefined.

Antibody, viral vector and subunit vaccines based on gB, the trimer and the pentamer are currently under development with a special interest in the prevention of congenital HCMV [23]. However, to date, these candidates have shown only a modest efficacy in precluding vertical transmission of HCMV [24]. Targeting cellular receptors for HCMV represents an untested alternative vaccine and therapeutic approach. The HCMV trimer mediates cell entry by binding to platelet-derived growth factor receptor alpha [25,26], and neuropilin-2 is the only bona fide cellular receptor identified to date for the HCMV pentamer [27]. Identification of the putative cellular 7-transmembrane domain receptors for CMV-encoded chemokines could provide new targets.

A third potential target for development of vaccines or therapeutics involves cellular glycosaminoglycans (GAGs), which mediate binding of many viruses to target cells [28]. In this regard, we have recently reported that like most human chemokines, MCMV MCK-2 binds directly to GAGs [29]. Although host chemokine interaction with GAGs is known to be essential for coordinating leukocyte trafficking in vivo [30], the functional significance of MCK-2-GAG interaction has not previously been investigated. GAGs are highly sulfated polysaccharides bound to the cell membrane through a protein core. Of the three major types of cell surface GAGs—heparan sulfate (HS), chondroitin sulfate (CS) and dermatan sulfate—HS appears to be the preferred target for most viruses, including HCMV [31,32,33,34,35,36,37]. This has led to the design of new anti-heparan sulfate peptides to mask cell surface anchor points for the virus [38]. Although they are very efficient in blocking infection by both HCMV and MCMV in vitro, these peptides are only partially protective in vivo [39,40]. Since most previous reports on the role of surface GAGs in viral entry studied a limited variety of target cell types (typically fibroblasts or epithelial cells in the case of HCMV) [34,41], lack of in vivo efficacy by anti-HS peptides could result if HCMV and MCMV interacted with cell type-specific GAGs differentially expressed on different types of target cells. In the present study, we addressed this possibility for MCMV. In addition, we investigated whether MCK-2-dependent infection of target cells by MCMV is influenced by the binding of MCK-2 to cell surface GAGs.

## 2. Materials and Methods

### 2.1. Animals

BALB/cJ mice were obtained from The Jackson Laboratory (Bar Harbor, ME, USA). All mice were maintained under specific pathogen-free housing conditions at an American Association for the Accreditation of Laboratory Animal Care-accredited animal facility at the National Institute of Allergy and Infectious Diseases (NIAID) and housed in accordance with the procedures outlined in the Guide for the Care and Use of Laboratory Animals under the protocol LMI-8E approved on 31 December 2015 and annually renewed by the Animal Care and Use Committee of NIAID.

### 2.2. Cells

Mouse fibroblast cell lines NIH-3T3 and M2-10B4 and the mouse macrophage cell line RAW264.7 were purchased from ATCC (Manassas, VA, USA). NIH-3T3 and RAW264.7 cells were maintained in DMEM-Glutamax (Life Technologies, Carlsbad, CA, USA) supplemented with 10% FBS. M2-10B4 cells were cultured in RPMI-Glutamax (Life Technologies, Carlsbad, CA, USA) supplemented with 10% FBS. Salivary gland- and lung-derived fibroblasts (SGFBs and LGFBs) were isolated from BALB/cJ mice as previously described [29]. Primary bone marrow-derived macrophages (BMDMs) were obtained from whole bone marrow cells extracted from the femurs and tibias of BALB/cJ mice and grown in RPMI-Glutamax media containing 20% FBS and 30% L929-conditioned media, as previously described [42]. Salivary gland acinar cells (SGACs) were cultured as previously described [43]. Briefly, salivary glands of BALB/cJ mice were minced and digested in DMEM/F12 media (Life Technologies, Carlsbad, CA, USA) containing 0.14 Wunsch units/mL of Liberase TL and 10 μg/mL DNAse I, both enzymes from Roche (Indianapolis, IN, USA), for 1 h at 37 °C. The digested tissue was filtered through 70 μm cell strainers, centrifuged at 200× *g* for 5 min, and then washed twice with PBS. SGACs obtained from 6 mice were resuspended in 10 mL of bronchial epithelial basal medium (Lonza, Atlanta, GA, USA) supplemented with bronchial epithelial growth medium bullet kit (Lonza, Atlanta, GA, USA) and plated in 96-well plates previously coated with 150 μg/mL of Cultrex Basement Membrane Extract (Trevigen, Gaithersburg, MD, USA). Cells were washed with PBS and media was replaced at 24 and 48 h after isolation. Cells were maintained in culture for three additional days before being used in experiments.

### 2.3. Viruses

The viruses MCMV-3D (MCK-2 knock out) and MCMV-3DR (wild type) were generously provided by Dr. Martin Messerle (Hannover, Germany). These viruses were engineered to express *Gaussia* luciferase and it was previously demonstrated that they establish productive infection in tissue culture and in mice [44,45]. Viruses were reconstituted from bacmids by transfection of NIH-3T3 cells and were expanded in M2-10B4 cells. Virus was collected by centrifugation (15,000× *g* for 3 h) of supernatants from 100% infected M2-10B4 cells, and then further purified by ultracentrifugation (20,000 rpm for 1 h) through a 15% sucrose cushion in virus standard buffer (VSB, 50 mM Tris-HCl pH 7.8, 12 mM KCl and 5mM EDTA). The pellet was resuspended in 0.5–1.0 mL of VSB, aliquoted and stored at −80 °C. Viral titers were determined by plaque assays in M2-10B4 cells as previously described [46].

### 2.4. Luciferase-Based Infectivity Assays

Cells seeded in a 96-well plate were infected at the indicated multiplicity of infection (moi) with MCMV-3D or MCMV-3DR in RPMI supplemented with 2% FBS. After a 2 h viral adsorption at 37 °C, the cells were washed once with PBS to remove unbound viral particles and then incubated in RPMI 10% FBS. The luciferase activity in 10 μL of supernatant was determined 18 h post infection (hpi) using the Biolux Gaussia Luciferase assay kit (New England Biolabs, Ipswich, MA, USA) and a Mithras LB 940 luminometer (Berthold, Oak Ridge, TN, USA). The relative luminescence units (RLUs) obtained in supernatants from mock-infected cells were subtracted from all samples during the analysis. To study the role of different cellular GAGs in the infectivity of MCMV, where indicated, cells were incubated prior to infection with 1 U/mL of heparinase II (HepII) or/and chondroitinase ABC (ChABC) (Sigma, St. Louis, MO, USA) for 30 min at 37 °C in the corresponding reaction buffer (HepII buffer: 20 mM Tris-HCl pH 7.5, 4mM CaCl_2_, 50 mM NaCl, and 0.01% BSA; ChABC buffer: 50 mM Tris-HCl pH 8.0, 60 mM sodium acetate, and 0.02% BSA). Subsequently, cells were washed once with PBS to remove the enzyme and infected with MCMV-3D or MCMV-3DR at the indicated moi in RPMI supplemented with 10% FBS. The luciferase activity in the supernatant was determined 18 hpi as explained above. Of note, we noticed that the ChABC buffer weakens the cell attachment to the plate, which may result in the loss of some cells during washes and final luciferase levels lower than in cells treated with the HepII buffer (See Figure 1B, panel SGFB).

### 2.5. Heparin-Binding Assays

To evaluate the presence of soluble GAGs in the culture media of M2-10B4 cells, we analyzed the capacity of M2-10B4-conditioned supernatants to block the binding of B18, a GAG-binding protein encoded by vaccinia virus [47], to heparin in an enzyme-linked immunosorbent assay (ELISA). For this, 10 µg/mL of biotinylated heparin (Sigma, St. Louis, MO, USA) was immobilized on streptavidin-coated ELISA plates (Pierce, Philadelphia, PA, USA) for 2 h at room temperature in assay buffer (Tris-buffered saline pH 7.2, 0.05% Tween-20 and 0.1% BSA). The cell-free supernatant from confluent M2-10B4 cells was treated with 1 U/mL of HepII or ChABC, or an equivalent volume of PBS for 40 min at 37 °C, then incubated 30 min at room temperature in a 1:1 vol ratio with 100 nM recombinant His-tagged B18 protein (generous gift from Dr. Antonio Alcami, Madrid, Spain) in assay buffer. Wells were washed 4 times with assay buffer and the B18:M2-10B4 supernatant solutions were added and incubated for 30 min at room temperature. Unbound B18 protein was removed by washing the wells 4 times with assay buffer and bound protein was detected with an anti-His mAb (Qiagen, Valencia, CA, USA) followed by an HRP-coupled anti-mouse secondary antibody (Santa Cruz Biotechnology, Santa Cruz, CA, USA) in assay buffer. Plates were developed with TMB One Component (Surmodics, Eden Prairie, MN, USA) and the reaction was stopped with sulfuric acid. The absorbance at 450 nm (A_450_) was measured in a FlexStation 3 microplate reader (Molecular Devices, Sunnyvale, CA, USA).

### 2.6. B18 Cell Surface Binding Assays

The binding of B18-His to the surface of M2-10B4, RAW264.7 and BMDM cells was analyzed by FACS. 3 × 10^5^ cells were incubated on ice for 20 min with buffer alone or increasing concentrations (5, 50 and 150 nM) of B18-His in PBS-staining buffer (PBS, 1% FBS and 1% BSA). Beforehand, cellular Fc receptors were blocked using TruStain FcX antibody solution (Biolegend, San Diego, CA, USA). After washing with PBS, cell-bound protein was detected with an anti-His mAb (Qiagen, Valencia, CA, USA) and an anti-mouse Alexa Fluor 488 secondary antibody (Life Technologies, Carlsbad, CA, USA). In total, 20,000 events were collected in a BD LSR Fortessa analyzer (BD Bioscience, San Jose, CA, USA), and the data were analyzed using FlowJo software (BD Bioscience, San Jose, CA, USA).

### 2.7. qPCR Analysis of Acinar Cell Markers

The relative expression of 4 common markers of salivary acinar cells—*Tjp1*, *Aqp5*, *Aqp3* and *Amy1*—was calculated in mouse salivary glands and cultured SGACs by qPCR. For this, the total RNA from whole submaxillary salivary glands of BALB/cJ mice and from 5 day cultures of SGACs was isolated with Trizol (Life Technologies, Carlsbad, CA, USA) following the manufacturer’s instructions. Genomic DNA was removed using the Turbo DNA-free kit (Life Technologies, Carlsbad, CA, USA). cDNA was synthesized from 500 ng of RNA using the SensiFast cDNA synthesis kit (Bioline, Cincinnati, OH, USA). Another 500 ng of RNA from each sample were mock treated without reverse transcriptase. The Cq values for the amplification of each acinar marker and the reference gene *Gapdh* were obtained in a CFX96 Real-Time System (Bio-Rad, Richmond, CA, USA) using the SensiFast SYBR kit (Bioline, Richmond, CA, USA) and the following primer pairs: Gapdh F (5′-aactttggcattgtggaagg-3′) and Gapdh R (5′-acacattgggggtaggaaca-3′); *Amy1* F (5′-gaaaagatgtcaatgactggg-3′) and *Amy1* R (5′-accatgttccttatttgacg-3′); *Aqp3* F (5′-cttctttgatcagttcataggc-3′) and *Aqp3* R (5′-gggttgttataagggtcaac-3′); *Aqp5* F (5′-atcttgtggggatctacttc-3′) and *Aqp5* R (5′-tagaagtagaggattgcagc-3′); *Tjp1* F (5′-ctgatagaaaggtctaaaggc-3′) and *Tjp1* R (5′-tgaaatgtcatctctttccg-3′). The relative expression for each marker was calculated as 2^(Cq^_marker −_
^Cq^_GAPDH_^)^.

### 2.8. MCMV Cell Binding Assays

The capacity of MCMV-3D and MCMV-3DR to bind to the surface of fibroblasts and macrophages was evaluated by qPCR or by reinfection of M2-10B4 cells with cell lysates. After purification, viral stocks used in qPCR-based assays were resuspended in PBS and treated overnight with 625 U/mL of benzonase (Sigma, St. Louis, MO, USA) at 4 °C to remove free DNA. Then, viral titers were determined by qPCR as the number of copies of the viral gene iE1/mL, interpolating in a standard curve generated with a pcDNA3.1-iE1 plasmid, the Cq values obtained with increasing volumes of the viral stocks as template and the primers iE1 2F (5′-catctcctgtcctgcaacct-3′) and iE1 2R (5′-cttgggctgctgttgattct-3′). Subsequently, increasing viral copies (10^4^–10^8^) of MCMV-3D and MCMV-3DR were incubated with NIH-3T3 and BMDM cells for 30 min on ice to prevent virus internalization. Cells were washed three times with cold PBS to remove unbound virus and DNA was isolated using QIAamp Blood kit (Qiagen, Valencia, CA, USA). Cell-retained viral iE1 copies were quantified by qPCR as explained above and represented relative to the *Gapdh* copies calculated by interpolation in a standard curve generated with a pcDNA3.1-Gapdh plasmid. Alternatively, MCMV-3D and MCMV-3DR were incubated at moi = 0.5 with M2-10B4, NIH-3T3 or RAW264.7 plated in 24-well plates. After a 30 min incubation on ice, cells were profusely washed with cold PBS and collected with a cell scraper in 100 µL of RPMI 2% FBS. Cells were lysed by three cycles of freeze and thaw and lysates were clarified by centrifugation. M2-10B4 cells in 96-well plates were infected with increasing volumes of the input stocks or 25 µL of the lysates and the luciferase activity in the supernatants was measured 18 hpi, as explained above. The RLUs upon infection with the lysates were represented relative to the RLUs obtained per µL of input used for the infection of M2-10B4.

## 3. Results

### 3.1. MCMV Infectivity Is Differentially Influenced by Target Cell Type-Specific GAGs in an MCK-2 Independent Manner

MCMV is known to infect both mouse fibroblasts and macrophages. Moreover, using a pSM3fr bacmid-derived MCMV system, it has been reported that MCMV infection of macrophages, but not fibroblasts, is promoted by MCK-2 [17]. Here, we confirmed this result using the previously characterized *Gaussia luciferase* reporter viruses MCMV-3D (MCK-2 knockout) and MCMV-3DR (MCK-2 wild type) [44], which allowed us to monitor infection by luminometry. In particular, as shown in Figure 1A, when tested at two different multiplicities of infection, MCK-2 deficiency did not affect the level of MCMV infectivity in either of the two mouse fibroblast cell lines that we tested, NIH-3T3 and M2-10B4 (Figure 1A). We obtained the same result using primary fibroblasts prepared from mouse salivary glands (SGFBs), which is a major site of MCMV replication in vivo (Figure 1B,—Enzyme). In contrast, the infectivity of the mouse macrophage cell line RAW264.7 as well as of BMDM was 3–5 times higher for the wild type virus MCMV-3DR than for the MCK-2 deficient virus MCMV-3D (Figure 1A,B,—Enzyme).

Next, we investigated whether MCK-2-dependent or -independent MCMV infectivity depended on cell surface GAGs. For this, we first tested the infectivity of both MCMV-3D and MCMV-3DR in fibroblasts and macrophages treated beforehand with the GAG lyases HepII and ChABC to deplete specific types of GAGs from the cell surface. HepII removes heparin and HS [48], and ChABC removes three major forms of CS: chondroitin 4-sulfate, dermatan sulfate and chondroitin 6-sulfate [49]. In the fibroblast cell line M2-10B4, we found that the treatment with HepII diminished infectivity for both MCMV-3D and MCMV-3DR viruses by ~50% (Figure 1B), whereas, in contrast, ChABC enhanced the infectivity of both viruses by ~2-fold (Figure 1B). Likewise, for primary SGFBs, HepII decreased the infectivity of both viruses by ~50%, whereas ChABC increased the infectivity of both viruses, although to a lesser extent than for the fibroblast cell line (Figure 1B). The infectivity pattern for primary lung-derived fibroblasts (LGFBs) pretreated with either HepII or ChABC before MCMV-3DR infection fit this same pattern (Figure 1C). Finally, we infected M2-10B4 and LGFBs after pretreatment with both GAG lyases together (HepII + ChABC) and observed an additive effect—in which, the MCMV-3DR infectivity of the cells was not different from the buffer control levels (Figure 1C).

To determine whether the presence of soluble GAGs in cultured M2-10B4 fibroblast supernatants might contribute to the effects of GAG lyases on MCMV infectivity, we tested the influence of GAG lyase-treated M2-10B4 cell-free supernatants on the binding of purified B18 protein to heparin (Figure 1D). B18 is an interferon-binding protein that is conserved in many poxviruses and that binds directly to GAGs, including HS and CS, with high affinity (in the nanomolar range) [47]. As shown in Figure 1D, a PBS-treated supernatant from cultured M2-10B4 fibroblasts (M2-10B4 SN, PBS) inhibited 60% of the B18-heparin binding observed when B18 was preincubated with fresh RPMI 10% FBS media (media), suggesting that soluble GAGs are released by M2-10B4 cells. Consistent with this interpretation, when the M2-10B4 supernatant was pretreated with ChABC (M2-10B4 SN, ChABC), the B18 binding to heparin increased by at least 20% (Figure 1D). HepII treatment of the supernatant had only a small effect on B18 binding to heparin. Therefore, the enhancement of MCK-2-independent MCMV infectivity in fibroblasts treated with ChABC might be explained in part by the depletion of soluble CS that may interfere with the binding of the virus to the cell surface. Consistent with this, it has been previously reported that less than 10% of the soluble GAGs produced by M2-10B4 cells correspond to HS [50].

The effect of GAG lyase treatment on MCMV infectivity in macrophages, however, fundamentally differed from our observations in fibroblasts and were different for the macrophage cell line RAW264.7 compared with primary BMDM. For RAW264.7 cells, neither HepII nor ChABC pretreatment affected the infectivity of either MCMV-3D or MCMV-3DR viruses (Figure 1B). This could be explained if GAGs were absent or scarce on the surface of these cells. Consistent with this, we observed that direct binding of the GAG-binding protein B18 to the surface of RAW264.7 cells was very low compared to the high B18 binding observed for M2-10B4 cells (Figure 1E). However, the possibility that MCMV may bind to other HepII- and ChABC-resistant cellular GAGs on RAW264.7 cells cannot be excluded. Yet a third pattern of GAG lyase-dependent MCMV infectivity was observed for primary BMDMs—in which, treatment of the cells with ChABC, but not with HepII, reduced the infectivity of both viruses, and to the same extent (~75% inhibition of control infectivity) (Figure 1B). In a separate set of experiments presented in Figure 1C, treatment with ChABC also reduced MCMV-3DR infectivity in BMDM, although to more modest levels (~40% inhibition), and we confirmed that HepII treatment does not affect MCMV infection of macrophages. Accordingly, treatment of BMDM with both enzymes combined did not decrease MCMV infectivity beyond the inhibition caused by ChABC alone (Figure 1C). In addition, the stronger binding of B18 to BMDM than to RAW264.7 cells suggests a higher abundance of surface GAGs in primary macrophages (Figure 1E). Together, these results suggest that the contribution of MCK-2 to viral replication in macrophages is not GAG-mediated and that CS on the surface of BMDM facilitates MCMV infection. Thus, MCMV appears to use different GAGs to initiate infection in different host cell types, using HS in fibroblasts but CS and potentially other GAGs in macrophages.

### 3.2. MCK-2 Is Not Required for the MCMV Infection of Mouse Salivary Gland Acinar Cells (SGACs)

As mentioned previously, MCK-2 is essential for MCMV infection of mouse salivary gland in vivo. A simple explanation for this would be that MCK-2, as a component of the glycoprotein entry complex gH/gL/MCK-2, might be required for the infection of SGACs. To test this hypothesis, we analyzed the infectivity of MCMV-3D and MCMV-3DR in explants of SGACs (Figure 2A). For this, we used a previously described method to isolate and grow acinar cells in vitro [43]. Before infection, we confirmed that, although to a lesser extent compared to the whole salivary gland, SGACs expressed common markers of acinar cells, including aquaporin-5 (*Aqp5*) and amylase-1 (*Amy1*) (Figure 2B). A reduction in the expression levels of these markers in cultured SGACs has been previously observed [43]. As shown in Figure 2C, we did not detect any difference in the infectivity of MCMV-3D and MCMV-3DR viruses in SGACs, suggesting that MCK-2 may be dispensable for the direct infection of acinar cells by MCMV. Additionally, we found yet a fourth pattern of GAG lyase-dependent MCMV infectivity in these cells: HepII, but not ChABC, significantly and equally reduced the infectivity of both viruses in these cells (Figure 2C), which indicates that HS facilitates MCMV infection of SGACs.

### 3.3. MCK-2 is Dispensable for MCMV Attachment to the Surface of Fibroblasts and Macrophages

Since the depletion of specific GAGs from the surface of fibroblasts, macrophages and SGACs altered MCMV infectivity in an MCK-2-independent manner, and since cellular GAGs are the first point of contact for the virus, we reasoned that MCK-2 may not be involved in MCMV cell tethering. To confirm this, we performed MCMV–target cell binding experiments with fibroblasts and macrophages using qPCR to quantify the viral copies of MCMV-3D and MCMV-3DR retained on the cell surface. As shown in Figure 3A, we did not observe any difference in the capacity of these viruses to bind to either NIH-3T3 cells or BMDM. Moreover, when lysates of M2-10B4, NIH-3T3 or RAW264.7 cells preincubated on ice with MCMV-3D or MCMV-3DR were used to infect M2-10B4 cells, which are equally susceptible to both viruses (Figure 1A), the luciferase activity detected after infection with MCMV-3D- or MCMV-3DR-treated lysates was similar in all cases (Figure 3B). Thus, MCK-2 is dispensable for MCMV binding to the cell surface and may regulate infectivity at a later stage of MCMV entry in macrophages.

## 4. Discussion

In the present study, we have demonstrated that MCMV infection of diverse mouse target cell types is markedly and differentially affected by cell surface GAGs. In particular, infectivity in primary mouse salivary gland- and lung-derived fibroblasts was inhibited by HepII pretreatment but enhanced by ChABC pretreatment. For primary salivary gland acinar cells, infectivity was reduced by HepII pretreatment but was unaffected by ChABC pretreatment, whereas the converse was observed for infectivity in primary mouse BMDM pretreated with these two GAG lyases. Cultured cell lines revealed two additional GAG lyase susceptibility patterns for MCMV infection: neither GAG lyase affected infectivity in the cultured mouse macrophage cell line RAW264.7, whereas, as for primary fibroblasts, the two enzymes oppositely affected infectivity in the cultured fibroblast cell line M2-10B4 (HepII pretreatment decreased whereas ChABC pretreatment increased infectivity). These results imply that HS and CS GAGs have differential target cell type-specific effects on MCMV infectivity, with HS being most important in supporting infection in primary fibroblasts and acinar cells and CS being most important in supporting infection in primary macrophages.

We have previously reported that the MCMV chemokine MCK-2, like host chemokines, can bind directly to GAGs [29], and others have reported that MCK-2 is a component of the MCMV trimer complex gH/gL/MCK-2 required for infection of macrophages in vitro [17]. Here, we have confirmed the importance of MCK-2 in macrophage infection, but we could not demonstrate that the mechanism involves interaction of MCK-2 with cellular GAGs. In HCMV, gB and the gM/gN complex have been identified as the main mediators of the initial attachment of the virus through cell surface GAGs [34,51]. Therefore, it is possible that their orthologues in MCMV play a similarly dominant role in the GAG-mediated binding of the virus to the cell surface. It also remains possible that MCK-2 mediates viral attachment through a GAG that is insensitive to HepII and ChABC. Of note, our data also do not preclude GAG regulation of the activity of MCK-2 as a soluble chemokine. As with human chemokines, the interaction of MCK-2 with GAGs may be essential for the formation of chemotactic gradients and the accumulation of this viral chemokine at the site of MCMV infection. The fact that the cellular receptor for MCK-2 remains unknown hinders to some extent our ability to explore these questions.

MCK-2-deficient MCMV reaches significantly lower viral titers specifically in the salivary gland when compared to an MCK-2-expressing virus [19,20]. However, the underlying mechanism responsible for this defect has not been defined. It has been reported that, in gH/gL/gO-competent MCMV, MCK-2 is dispensable for the infection of the salivary gland in immunocompromised mice [52]. Consistent with this, here we show that the direct infection of MCMV of SGFBs and SGACs, the main virus-producing cells in the mouse salivary gland, is unaffected in the absence of MCK-2. As an alternative possibility, the contribution of MCK-2 to MCMV pathogenesis in the salivary gland of immunocompetent mice might be related to a possible immunomodulatory activity that may impede the clearance of the infection in this organ, or to a heightened infection of other cell types, like monocytes or macrophages, that may support viral dissemination to the salivary gland.

The cell type-dependent GAG effects on MCMV infection reported here might be explained by differential availability of GAGs in different target cells. In fact, consistent with our results, CS is usually the most abundant type of GAG in macrophages (55–77% of the total GAG content) [53,54,55]. More specifically, the cell surface of BMDM from BALB/cJ mice was reported to contain 75% CS and 25% HS [53]. In contrast, in fibroblasts, endothelial and epithelial cells, HS typically comprises 50–80% of the surface GAG content [56,57,58,59,60,61]. This may also explain why previous studies limited to fibroblasts or epithelial cells identified HS as the main type of GAG exploited by HCMV and other viruses to bind to the target cell. Numerous factors might determine to what extent GAGs influence infectivity, but the relevant abundance of different virus-interacting GAGs on a given target cell and their relative binding affinity for the virus would logically play dominant roles.

Anti-HS peptides have been engineered to block HCMV and MCMV cell attachment [38]. However, these peptides displayed only a partial anti-MCMV activity in vivo [39,40]. Yet, in vitro, anti-HS peptides excelled in blocking the infection of MCMV in NIH-3T3 fibroblasts and mouse embryonic fibroblasts [38,40,62], and of HCMV in human lung embryonic fibroblasts, iris stroma cells, foreskin fibroblasts, aorta endothelial cells, endothelial HUVEC cells, epithelial ARPE-19 cells and MRC-5 fibroblasts [38,39,40,41,62]. Coincidentally, all of these cells are either fibroblasts, endothelial or epithelial cells, and therefore, they are likely to present a surface GAG-profile dominated by HS. The anti-viral activity of these peptides in HS-deficient MCMV target cells has not been tested yet; however, our results suggest they might be ineffective in macrophages. Macrophages, monocytes, neutrophils and dendritic cells have been reported to be some of the cell types that MCMV infects first and hijacks for its dissemination to distal organs [63,64,65,66]. In these cell types, CS is the most abundant surface GAG, and their HS levels range from low (macrophages, monocytes and dendritic cells) to non-detectable (neutrophils) [53,67,68], which suggests that anti-HS peptides would have a poor anti-MCMV effect in these cells. The inability of anti-HS peptides to block MCMV entry in these myeloid cells and the subsequent cell-to-cell spread of the virus, could explain the limited protective activity of these agents in vivo. A mix of anti-HS and anti-CS peptides might prove to be more effective since they would target a wider spectrum of MCMV-infectable host cells. Our data may open up new possibilities for GAG regulation of viral infection that might be explored for other viruses and their target cells. Advances in cell glycobiology might significantly help to more fully understand the role of GAGs in viral infection. These lines of research might reveal new targets for the design of new anti-viral compounds to block virus–target cell interactions.

## Figures and Tables

**Figure 1 viruses-12-00031-f001:**
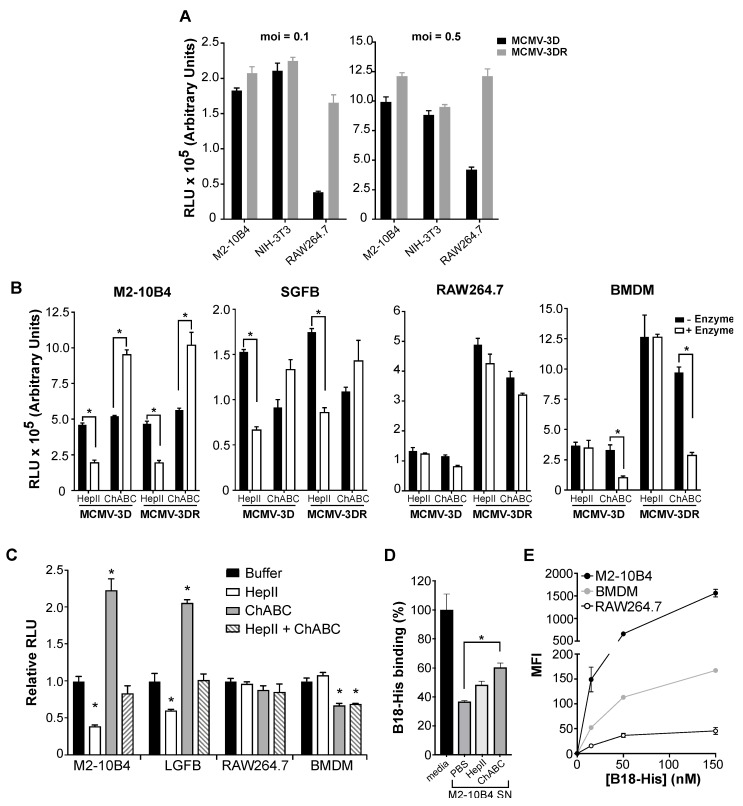
Mouse cytomegalovirus (MCMV) infection of diverse target cell types is differentially regulated by glycosaminoglycans (GAGs) in an MCK-2-independent manner. (**A**) MCK-2 deficiency impairs the MCMV infectivity of macrophages but not fibroblasts. Infectivity was defined as relative luminescence units (RLUs) recorded 18 hpi in 10 µL of supernatant from the target cells indicated on the x-axis infected at the moi indicated above each graph with the *Gaussia* luciferase reporter viruses MCMV-3D (MCK-2 deficient MCMV, black bars) or MCMV-3DR (MCK-2 expressing wild-type MCMV, gray bars). M2-10B4 and NIH-3T3 are mouse fibroblast cell lines; RAW264.7 is a mouse macrophage cell line. (**B**,**C**) GAGs differentially affect target cell infectivity by MCMV. In B, the cell types indicated above each graph (SGFBs, primary mouse salivary gland fibroblasts; BMDMs, primary mouse bone marrow-derived macrophages) were pretreated with the GAG lyases indicated on the x-axis (HepII, heparinase II; ChABC, chondroitinase ABC), and then infected with viruses (moi = 0.1) as indicated below the x-axis. White bars, + Enzyme; black bars, - Enzyme (reaction buffer without enzyme). Infectivity was determined at 18 hpi as in panel A. In panel C, the effect of the GAG lyase treatments indicated in the inset on the infectivity of MCMV-3DR (moi = 0.1) for the target cells indicated on the x-axis (LGFBs, primary mouse lung fibroblasts) were analyzed as in panel B. Results are shown relative to the RLUs detected in the “Buffer” group (reaction buffer without enzyme) for each target cell type. (**D**) The MCMV infectivity of mouse M2-10B4 fibroblasts may be restricted by soluble GAGs. Binding of His-tagged recombinant vaccinia protein B18 to heparin was defined by enzyme-linked immunosorbent assay (ELISA). Activity was measured after incubation of B18-His with the materials indicated on the x-axis. Media, RPMI 10% media; M2-10B4 SN, supernatant from cultured mouse fibroblast cell line M2-10B4 previously treated with PBS, or with the GAG lyases HepII or ChABC. Data are presented as the absorbance at 450 nm (A_450_) relative to the A_450_ recorded in wells containing B18-His incubated with fresh media which was defined as 100%. (**E**) Differential levels of cell surface GAG expression on MCMV target cells. GAG expression was defined by FACS by binding of B18-His to the cell surface. The cell types are indicated in the inset. Data are presented as the median fluorescence intensity (MFI) calculated after subtraction of the MFI recorded on cells incubated with buffer alone. (**A**–**E**) In all experiments, data are represented as the mean ± SD of triplicates from one experiment representative of three to four independent experiments. Results in B and C were analyzed by multiple *t* tests and by one-way ANOVA in D. (*, *p* < 0.05).

**Figure 2 viruses-12-00031-f002:**
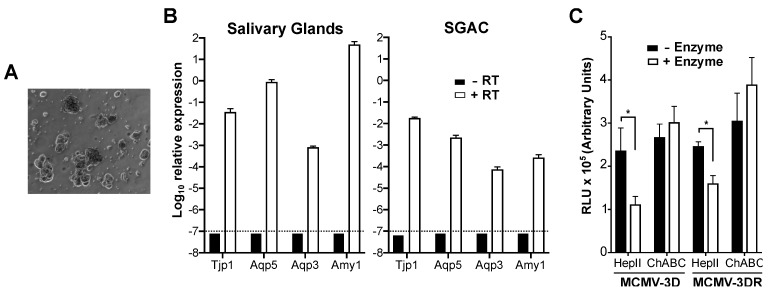
MCMV infection of mouse salivary gland acinar cells is independent of MCK-2 and promoted by cell surface heparan sulfate. (**A**) Appearance of cultured purified primary salivary gland acinar cells (SGACs). A representative 20× bright field image is shown one day after isolation. (**B**) Purified SGACs maintain their molecular phenotype in culture. qPCR analysis of the relative expression of common SGAC markers (*Tjp1*, tight junction protein 1; *Aqp5*, aquaporin 5; *Aqp3*, aquaporin 3; *Amy1*, amylase 1) in whole mouse salivary glands or cultured SGACs (as indicated above each graph). The expression of each gene relative to the expression of *Gapdh* in retrotranscribed (+ RT, white bars) or non-retrotranscribed (-RT, black bars) RNA samples is indicated. The dashed horizontal lines mark the detection limit. Data are represented as the mean ± SD of triplicates from one experiment representative of two independent experiments. (**C**) MCMV infection of SGACs. 5 days after isolation, SGACs pretreated (+ Enzyme, white bars) or not (-Enzyme, black bars) with the GAG lyases (HepII or ChABC) indicated on the x-axis were infected with 10,000 pfu of the *Gaussia* luciferase reporter viruses MCMV-3D or MCMV-3DR, and the RLUs in 10 µL of supernatant was determined 18 hpi by luminometry. Data are presented as the mean ± SD of triplicates from one experiment representative of three independent experiments. Statistically significant differences were determined by multiple *t* tests (*, *p* < 0.05).

**Figure 3 viruses-12-00031-f003:**
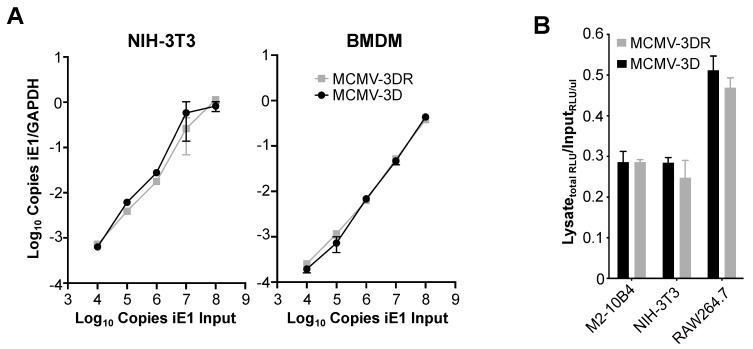
MCK-2 is dispensable for MCMV binding to the surface of fibroblasts and macrophages. (**A**) Molecular analysis. The cell types indicated above each graph were incubated on ice with increasing copy numbers (10^4^–10^8^) of the MCMV viruses indicated by the inset in the right panel. Virus copy numbers were quantitated as the number of viral *iE1* gene copies (input) in the corresponding viral stocks. After extensive PBS washing to remove unbound virus, total DNA was isolated and the total copies of cellular *Gapdh* and viral *iE1* were determined by qPCR. Data are presented as the mean ± SD of triplicate determinations of the ratio of (copies iE1)/(copies Gapdh) from one experiment and are representative of three independent experiments. (**B**) Infectivity analysis. The cell lines indicated on the x-axis were incubated at an moi = 0.5 on ice with the *Gaussia* luciferase reporter viruses MCMV-3D (black bars) or MCMV-3DR (gray bars). After several PBS washes to remove unbound virus, cells were collected and lysed by three freeze-thaw cycles. Fresh M2-10B4 cells were infected with viral inputs or cell lysates and 18 hpi the relative luminescence units (RLUs) in 10 µL of supernatant was determined. Data are quantitated as the total RLUs obtained after infection with the cellular lysates relative to the RLUs obtained per µL of viral input used to infect M2-10B4 cells. Data are presented as the mean ± SD of triplicate determinations from one experiment and are representative of three independent experiments. M2-10B4 and NIH-3T3 are mouse fibroblast cell lines; RAW264.7 is a mouse macrophage cell line.

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
