# Peer review of "Mouse Cytomegalovirus Differentially Exploits Cell Surface Glycosaminoglycans in a Cell Type-Dependent and MCK-2-Independent Manner"

_viruses, 2019, doi:10.3390/v12010031_

Round 1

Reviewer 1 Report

In this study, Pontejo and Murphy investigated the requirement of cell surface glycosaminoglycans (GAG) in MCMV infection of different cell types. The authors showed that MCMV targets heparan sulfate (HS) in fibroblast and salivary gland acinar cells (SGAC) but it mainly targets chondroitin sulfate in macrophage cell lines and bone-marrow derived macrophages. MCMV-encoded MCK2, a GAG binding protein, promotes the infection of macrophages but is dispensable for MCMV binding to the cell surface and direct infection of SGAC. The authors conclude that MCMV tropism is significantly influenced by GAG expression.

Specific comments;

Page 6 Line 243-244, the relevant references for the sentence should be cited. “HepII removes heparin and HS, and ChABC removes 3 major forms of CS: chondroitin 4-sulfate, dermatan sulfate and chondroitin 6-sulfate” Fig 1B and Fig 2B, the treatment of cells with combination of HepII and ChABC should be used to examine the MCMV infection in the absence of more types of GAGs.

Author Response

Response to Reviewer 1 Comments
In this study, Pontejo and Murphy investigated the requirement of cell surface
glycosaminoglycans (GAG) in MCMV infection of different cell types. The authors showed that MCMV targets heparan sulfate (HS) in fibroblast and salivary gland acinar cells (SGAC) but it mainly targets chondroitin sulfate in macrophage cell lines and bone-marrow derived macrophages. MCMV-encoded MCK2, a GAG binding protein, promotes the infection of macrophages but is dispensable for MCMV binding to the cell surface and direct infection of SGAC. The authors conclude that MCMV tropism is significantly influenced by GAG expression.

Point 1: Page 6 Line 243-244, the relevant references for the sentence should be cited. “HepII removes heparin and HS, and ChABC removes 3 major forms of CS: chondroitin 4-sulfate, dermatan sulfate and chondroitin 6-sulfate”

Response 1: The requested references have been added. Page 6, lines 256 and 257 of the revised manuscript.

Point 2: Fig 1B and Fig 2B, the treatment of cells with combination of HepII and ChABC should be used to examine the MCMV infection in the absence of more types of GAGs.

Response 2: We have conducted new experiments and added a new Figure 1C showing the effect of combined HepII/ChABC pretreatment on MCMV infectivity of target cells. We found an additive effect in fibroblasts but not in macrophages.

Reviewer 2 Report

Pontejo and Murphy provide strong evidence that MCMV binds to heparan sulfate to infect fibroblasts and cultured salivary gland acinar cells, and chondroitin sulfate to bind to macrophage cell lines and bone-marrow derived macrophages. They also demonstrate that the viral GAG-binding chemokine MCK-2, which is part of the ternary gH/gL/MCK-2 complex that mediates entry into epithelial and endothelial cells, is dispensable for interaction with the cell surface and infection of salivary gland acinar cells. Finally, they show that MCK-2 is dispensable for binding to the surface of fibroblasts and macrophages. 

Overall, this is a very well-executed study and a well-written manuscript. The authors methodically addressed the role of MCK-2 in MCMV entry using an available mutant virus and revertant. One of the notable strengths of the study was the use of a variety of primary cell types and cultured cell lines to methodically investigate the role of HS and CS GAGs and MCK-2 in binding and entry. The use of the B18-His fusion protein to investigate the role of soluble GAGs in regulating binding was particularly innovative. The study could be further strengthened by providing clear demonstration of the activity of the GAG-degrading enzyme preparations HepII and ChABC, which could be measured directly by HPLC. However, the absence of such analysis does not significantly detract from an excellent study. 

Author Response

Response to Reviewer 2 Comments
Pontejo and Murphy provide strong evidence that MCMV binds to heparan sulfate to infect fibroblasts and cultured salivary gland acinar cells, and chondroitin sulfate to bind to macrophage cell lines and bone-marrow derived macrophages. They also demonstrate that the viral GAG-binding chemokine MCK-2, which is part of the ternary gH/gL/MCK-2 complex that mediates entry into epithelial and endothelial cells, is dispensable for interaction with the cell surface and infection of salivary gland acinar cells. Finally, they show that MCK-2 is dispensable for binding to the surface of fibroblasts and macrophages.

Point 1: Overall, this is a very well-executed study and a well-written manuscript. The authors methodically addressed the role of MCK-2 in MCMV entry using an available mutant virus and revertant. One of the notable strengths of the study was the use of a variety of primary cell types and cultured cell lines to methodically investigate the role of HS and CS GAGs and MCK-2 in binding and entry. The use of the B18-His fusion protein to investigate the role of soluble GAGs in regulating binding was particularly innovative.

Response 1: We thank the Reviewer for these comments.

Point 2: The study could be further strengthened by providing clear demonstration of the activity of the GAG-degrading enzyme preparations HepII and ChABC, which could be measured directly by HPLC. However, the absence of such analysis does not significantly detract from an excellent study.

Response 2: The GAG-degrading enzymes used in this study are quality-controlled commercially available GAG-lyases sold by units (Vendor’s unit definition: For heparinase II, One unit will form 0.1 μmole of unsaturated uronic acid per hr at pH 7.0 at 25 °C; For chondroitinase ABC, One unit will liberate 1.0 μmole of 2-acetamido-2-deoxy-3-O-(β-Dgluc- 4-ene-pyranosyluronic acid)-4-O-sulfo-D-galactose from chondroitin sulfate A or 1.0 μmole of 2-acetamido-2-deoxy-3-O-(β-D-gluc-4-ene-pyranosyluronic acid)-6-O-sulfo-Dgalactose from chondroitin sulfate C per min at pH 8.0 at 37 °C.) We have not repeated this quality control of the enzymes obtained from the vendor, but the reviewer is not requiring that we do so.

Round 2

Reviewer 1 Report

The authors addressed my concerns.